# Low-Malignant-Potential Adenocarcinoma: A Histological Category with a Significantly Better Prognosis than Other Solid Adenocarcinomas at IA Stage

**DOI:** 10.3390/curroncol32040217

**Published:** 2025-04-09

**Authors:** Marco Chiappetta, Alessandra Cancellieri, Filippo Lococo, Elisa Meacci, Carolina Sassorossi, Maria Teresa Congedo, Qianqian Zhang, Diomira Tabacco, Isabella Sperduti, Stefano Margaritora

**Affiliations:** 1Thoracic Surgery Unit, University “Magna Graecia”, 88100 Catanzaro, Italy; marcokiaps@hotmail.it; 2UOC di Chirurgia Toracica, Fondazione Policlinico Universitario A. Gemelli—IRCCS, 00168 Rome, Italy; filippo.lococo@policlinicogemelli.it (F.L.); elisa.meacci@unicatt.it (E.M.); mariateresa.congedo@policlinicogemelli.it (M.T.C.); stefano.margaritora@policlinicogemelli.it (S.M.); 3Division of Anatomic Pathology and Histology, Fondazione Policlinico Universitario A. Gemelli—IRCCS, 00168 Rome, Italy; alessandra.cancellieri@policlinicogemelli.it (A.C.); qianqian.zhang@policlinicogemelli.it (Q.Z.); 4Thoracic Surgery, UOC di Chirurgia Toracica, Università Cattolica del Sacro Cuore, 00168 Rome, Italy; 5UOC Chirurgia Toracica, Azienda Ospedaliero-Universitario Policlinico-San Marco, 95123 Catania, Italy; diomira_tabacco@yahoo.it; 6Biostatistical Unit, IRCCS Regina Elena National Cancer Institute, 00144 Rome, Italy; isabella.sperduti@ifo.it

**Keywords:** adenocarcinoma, NSCLC, low-malignant-potential adenocarcinoma

## Abstract

Introduction: Low-malignant-potential adenocarcinoma has been defined as a type of non-mucinous tumor, which has a total tumor size measuring ≤ 3 cm, exhibits ≥ 15% lepidic growth, lacks non-predominant high-grade patterns (≥10% cribriform, ≥5% micropapillary, ≥5% solid), has an absence of angiolymphatic or visceral pleural invasion, spread through air spaces (STAS), necrosis and >1 mitosis per 2 mm^2^. The aim of this study is to validate, with regard to cancer-specific survival (CSS) and disease-free survival (DFS), the proposed definition of LMP adenocarcinoma in an independent external cohort of lung adenocarcinoma patients having undergone surgical resection, and having presented with a long follow-up period. Methods: Clinicopathological characteristics of patients who underwent lung resection for adenocarcinoma from 1 January 2005 to 31 December 2014 were retrospectively analyzed. Patients with ground-glass opacity (GGO) and part-solid tumors, minimally invasive adenocarcinoma (MIA), adenocarcinoma in situ (AIS), tumors ≥5 cm in size, nodal involvement and/or distant metastases, patients who underwent neoadjuvant treatment, and those who had an incomplete follow-up or a follow-up shorter than 60 months were excluded. The proposed criteria for low-malignant-potential adenocarcinoma (LMPA) were tumor size ≤ 3 cm, invasive size ≥ 0,5 cm, lepidic growth ≥ 15%, and absence of the following: mitosis (>1 per 2 mm^2^), mucinous subtype, angiolymphatic invasion, visceral pleural invasion, spread through air spaces (STAS) and tumor necrosis. End points were disease-free survival (DFS) and cancer-specific survival (CSS). The log-rank test was used to assess differences between subgroups. Results: Out of 80 patients meeting the proposed criteria, 14 (17.5%) had the LMPA characteristics defined. The mean follow-up time was 67 ± 39 months. A total of 19 patients died, all in the non-LMPA category, and 33 patients experienced recurrence: 4 (28.5%) with LMPA and 29 (43.9%) with non-LMPA. Log-rank analysis showed 100% 10-year CSS for patients with LMPA and 77.4% for patients without LMPA, with this difference being statistically significant (*p*-value = 0.047). Univariate analysis showed a significant association with the cStage (AJCC eighth edition), both for CSS (*p* value = 0.005) and DFS (*p*-value = 0.003). LMPA classification did not show a statistically significant impact on CSS and DFS, likely due to the limited number of events (CSS *p*-value = 0.232 and DFS *p*-value = 0.213). No statistical association was found for CSS and DFS with pT, the number of resected nodes (< or >10) or the number of resected N2 stations (< or >2). Conclusions: Our study confirmed the prognostic role of LMPA features, with a low risk of recurrence and a good CSS and DFS. The criteria for diagnosis are replicable and feasible for application. The clinical implications of these findings, such as pre-operative prediction and surveillance scheduling, may be the topic of future prospective studies.

## 1. Introduction

Lung cancer is the leading cause of cancer-related death in both sexes (18% of total cancer deaths), with a progressive increment in cases worldwide in 2020, representing 11.4% of all new cancer cases [1,2]. Among different histologies, adenocarcinoma is the most common cancer subtype, accounting for 50% of all lung cancer diagnoses [2], and presents different subtypes with particular pathological characteristics, reflecting different prognosis. In 2011, the International Association for the Study of Lung Cancer (IASLC), the American Thoracic Society (ATS) and the European Respiratory Society (ERS) [3] proposed an international standard for histologic subclassification of lung adenocarcinoma, adopted in 2015 by the World Health Organization (WHO) [4]. In particular, the major changes in lung adenocarcinoma classification were the discontinuation of the use of the term bronchoalveolar carcinoma (BAC), and the introduction of adenocarcinoma in situ (AIS) and minimally invasive adenocarcinoma (MIA). AIS is defined as a solitary, small (≤3 cm) adenocarcinoma with pure lepidic growth. MIA is a small (≤3 cm), solitary adenocarcinoma with predominant lepidic growth and small foci of invasion measuring ≤ 0.5 cm.

It is important to note that these two adenocarcinoma spectrum lesions present, when radically resected, a 5-year disease-specific survival of 100% [5,6]. Moreover, a correlation between CT appearance and adenocarcinoma spectrum tumors has already been validated, with lesions beginning with a radiological appearance as ground-glass opacities, and then evolving to subsolid and solid lesions [7,8]. Therefore, four stages of lung adenocarcinoma development have been identified, with a continuum between the entities. The earliest form is adenomatous atypical hyperplasia (AAH), which progresses to AIS, then to MIA, and finally to a fully invasive adenocarcinoma [9]. However, this excellent outcome is reported for subsolid lesions, while the identification of prognostic groups in full solid tumors is still under definition. In 2020, Yambayev et al. [10] combined independently established histologic features to define a subgroup of adenocarcinoma, to be termed “low-malignant-potential” (LMP) adenocarcinoma, reporting an excellent prognosis with a 100% disease-specific survival. The LMP group was defined among non-mucinous tumors, with a total tumor size measuring ≤ 3 cm, exhibiting ≥ 15% lepidic growth, lacking non-predominant high-grade patterns (≥10% cribriform, ≥5% micropapillary, ≥5% solid), and with the absence of angiolymphatic or visceral pleural invasion, spread through air spaces (STAS), necrosis and >1 mitosis per 2 mm^2^. Knowledge of the clinical outcome of this particular subtype of adenocarcinoma may change the management of this cohort of patients. For example, limited lung resections may be planned for this subgroup of adenocarcinoma, followed by an appropriate surveillance program.

The aim of this study is to validate, with regard to cancer-specific survival (CSS) and disease-free survival (DFS), the proposed definition of LMP in an external cohort of lung adenocarcinoma patients having undergone surgical resection, with a long follow-up period.

## 2. Materials and Methods

The study was approved by the Ethics Committee of Policlinico Universitario A. Gemelli IRCCS N.0015316/22 ID4929 on 3 May 2022. Written informed consent was obtained from all the patients.

The clinical and pathological characteristics of patients who underwent lobectomy with a diagnosis of adenocarcinoma from January 2005 to December 2014 were retrieved from the files of Fondazione Policlinico Gemelli, Rome, and retrospectively analyzed. Patients were selected with the aim of obtaining a long follow-up of a minimum of 60 months, in the absence of end point events.

The inclusion and exclusion criteria were defined as follows:

Inclusion criteria

−Adenocarcinoma histology−Availability of pathological specimens for review−Tumor dimension < 3 cm−Lymphadenectomy−Complete resection−Complete follow-up information−Follow-up >60 month in patients without recurrence−Biopsy-proven recurrence

Exclusion criteria:−Neoadjuvant therapy−Adenocarcinoma in situ (AIS)−Minimally invasive adenocarcinoma (MIA)−Mixed adenocarcinoma histology−Nodal involvement or distant metastases−Cause of death not specified

All patients included underwent pre-operative assessment with thorax–abdomen computed tomography with contrast and, as has been the standard approach since 2009, 18-fluorodeoxyglucose positron emission tomography. After evaluation of lung function, surgery was performed, via thoracotomy or a video-assisted approach, by certified thoracic surgeons.

Pathological sections were separately reviewed by two pathologists, both experts in the field of lung pathology and blinded to follow-up, to identify cases that met the criteria for a diagnosis of LMPA. Discordant diagnoses were discussed and a diagnosis agreed upon unanimously was selected.

LMPA is defined by Yambayev et al. [8] as meeting the following criteria:−Size ≤ 3 cm;−Invasive size ≥ 0.5 cm;−Lepidic growth ≥ 15%;−High-grade components ≤ 5% (solid, micropapillary, cribriform);−Mitosis < 1 per 2 mm^2^;−Non-mucinous subtype;−Absence of angiolymphatic invasion;−Absence of visceral pleural invasion;−Absence of tumor necrosis;−Absence of spread through air spaces (STAS).

Patients meeting these pathological criteria were included in the LMPA group, while other patients with a tumor dimension ≤ 3, but not meeting all the criteria, were included in the non-LMPA group.

Post-operative follow-up consisted of clinical examination, a blood marker evaluation and a thorax–abdomen CT scan every 6 months for 5 years. In the case of suspected recurrence, a biopsy was attempted to verify the hypothesis.

### Statistical Analysis

Descriptive statistics were used to summarize pertinent study information. The association between variables was tested by the Pearson Chi-square test or Fisher’s exact test, when appropriate. The hazard ratio and confidence limits were estimated for each variable of interest’s outcome, using the Cox univariate model. Significance was defined at the *p* < 0.05 level. In the univariable analysis, baseline demographic, clinical and pathological characteristics, such as sex, age, pathological stage, #RN, #RS and LMPA histology, were considered. Disease-free survival (DFS) was calculated by the Kaplan–Meier product-limit method from the date of the surgery until relapse or death. If a patient had no relapse, DFS was censored at the time of the last visit. Cancer-specific survival (CSS) was calculated from the time of surgery until death due to cancer progression. Patients were classified as free from disease when medical examination and follow-up examinations were negative for suspected relapses or metastasis. The log-rank test was used to assess differences between subgroups. SPSS (version 29.0; SPSS, Inc., Chicago, IL, USA), a licensed statistical program, was used for all analyses.

## 3. Results

The final analysis was conducted on 80 patients meeting the inclusion criteria. Their clinical and pathological characteristics are reported in Table 1.

Most patients underwent lobectomy and nodal sampling. The mean number of resected lymph nodes was 8.4 ± 6.9, and the mean number of mediastinal lymph nodes was 4.5 ± 5.4.

Fourteen LMPAs (10M, 4F) (17.5%) were found. The mean age of the 14 patients was 73 years (for males, 73; for females, 72), with a median of 74 years. In all but one patient, co-morbidities were found, including ulcerative colitis, diabetes, dyslipidemia, Steven-Johnson’s syndrome, rheumatoid arthritis and other collagenous vascular diseases, systemic hypertension and COPD. Eleven patients (78.6%) were former or current smokers.

Patients did not meet the LMPA inclusion criteria in 21 cases (26.3%) because of the presence of STAS, which, in 10 cases, was also concurrent with features suggesting likely aggressive tumor behavior, necrosis and/or a consistent (>5%) component of high-grade histotypes; 18 (22.5%) patients were excluded for the presence of necrosis, and 27 additional cases were excluded from the LMP group because of the presence of a scarce or absent in situ component or a high grade > 5%.

### Survival Outcome

With a mean follow-up of 67 ± 39 months, 33 (41.3%) patients experienced a recurrence, 4 in the LMPA group (28%) and 29 (43.9%) in the non-LMPA group. The four recurrences recorded in the LMPA group occurred at 123, 60, 5 and 81 months, respectively, and of these, three recurrences were local (lung parenchyma) and surgically treated, while one was nodal in the laterocervical nodes, treated with chemotherapy. In the non-LMPA group, 15 recurrences were parenchymal, but only 1 was limited and surgically treated; 4 were in the mediastinal lymph nodes, and the remaining 10 involved distant structures such as brain, bones, liver and adrenal gland. All these other recurrences were treated with radiotherapy, chemotherapy or immunotherapy.

During follow-up, 19 patients died, of whom 14 died due to cancer progression, all in the non-LMPA group.

The Kaplan–Meier curve shows a significant CSS difference concerning LMPA status: the 10-year CSS for patients with LMPA was 100%, and it was 77.4% for patients without LMPA (*p*-value 0.047) (Figure 1).

The results from the univariate analysis are reported in Table 2. We found a significant association of the cStage (AJCC eighth edition) with both CSS (*p* 0.005) and DFS (*p* 0.003). In the univariate analysis, the presence of low-malignant-potential adenocarcinoma did not show a statistically significant association with either cancer-specific survival or disease-specific survival (for CSS, *p*-value = 0.232, and for DFS, *p*-value = 0.213, respectively). Furthermore, no statistical association was found for CSS and DFS with pT dimension, the number of resected nodes (< or >10) or the number of resected N2 stations (< or >2). No statistically significant difference was found for 10-year DFS with respect to the presence or absence of the LMPA, which reached 75% and 47.1%, respectively (*p*-value 0.20) (Figure 2).

Regarding the kind of recurrence (local or distant), no statistically significant difference was found between patients with or without LMPA (*p*-value = 0.847) (Table 3).

## 4. Discussion

In the present study, we aimed to validate the implications of low-malignant-potential adenocarcinoma with respect to prognosis.

In 2020, Yambayev et al. [10] analyzed independently established histologic features to define a subgroup of adenocarcinoma, defined as “low-malignant-potential” adenocarcinoma. In their report, LMP showed an excellent prognosis, with 100% DFS.

The survival outcomes for LMPA have not been comprehensively validated as of yet, and so no clinical implications have been defined.

In our cohort, after selection, 80 patients were included in the study. Among these, we identified 14 cases (17.5%) which met the proposed criteria for LMP adenocarcinoma. Our incidence is in line with a few previous studies that reported a rate of LMPA among other stage IA adenocarcinomas of 16% (Yambayev et al. [10]) and 12.4% (Pittaro et al. [11]).

Similarly, the resulting CSS is concordant with the findings of Yambayev et al. [10], and close to the results obtained by Pittaro and colleagues [11], as they obtained a disease-specific survival (DSS) of 94.1% in their validation study.

The authors [11] included 274 stage IA adenocarcinoma patients, of whom 34 (12.4%) met the proposed criteria, with a mean follow-up of 5.7 yrs. In their series, five LMPA patients (14.7%) had a recurrence of the disease, and two died from it. One of the recurrences was found 7 years after surgery. Our proportion of recurrence was slightly higher (28%, 4 patients out of 14) compared to that found by Pittaro et al. [11]. This difference may be due to the longer period of recruitment we had, from 2005 to 2014. Furthermore, no patients with LMPA died from disease progression in our cohort. Concerning the recurrence, Pittaro et al. suggest, as a conclusion of their work, a longer follow-up of up to 10 years, because of the very slow progression of the disease, and our results confirm these suggestions, in light of recurrences occurring after more than 60 months in three cases. Pittaro et al. also considered the need for ancillary techniques, such as Ki67, PHH3 and elastic fiber stain, which could be helpful for confirming the morphologic findings for LMPA diagnosis. Therefore, they focused their analysis on post-operative findings to confirm diagnosis, and on surveillance after surgery.

Patients with LMPA presented a 10-year DFS of 75%, compared to 47% in the non-LMPA group, with an improvement of about 30%, although this was not statistically significant. The non-significance of this result may be due to paucity of events and patients in our LMPA group, and future larger studies may be able to clarify the significant difference between the two groups.

The results from the univariate analysis instead demonstrated a statistically significant association of CSS and DFS only with pStage (according to the 8th edition of TNM), which is already known to be associated with prognosis [12].

The absence of significance for CSS in the univariate analysis was due to the absence of death in the LMP group, with a significant CSS difference confirmed in the log-rank test and with the Kaplan–Meier curve.

Considering these results, LMPA can be predictive of a lower risk of recurrence and of improved CSS and DFS. Consequently, it should be taken into account when planning surveillance and treatment strategies. Among those who have an adenocarcinoma smaller than 3 cm and are node-negative, some surveillance and therapeutic strategies have already been studied, according to post-operative (adenocarcinoma subgroup) and pre-operative features (radiological and metabolic). Indeed, different studies [13,14,15] have looked at prognostic factors influencing the prognosis of this group, but none of them have considered LMPA criteria.

In particular, Hung and coworkers [13] made a stratification for adenocarcinoma subgroups, and proposed that for patients with a predominantly micropapillary/solid pattern should be offered adjuvant chemotherapy, as they are more likely to have recurrence.

Kagimoto and colleagues [14] instead used metabolic features to predict patients that were candidates for sublobar resection. They used the Deauville scale, a five-point scale evaluation of FDG accumulation on PET/CT. Lung adenocarcinoma patients with a whole tumor size of 3 cm or smaller and a Deauville score of 1 or 2, according to their findings, could be candidates for sublobar resection.

Liu et al. [15] evaluated radiological features. They compared non-cavitary lung cancers and cavitary lung cancers smaller than 3 cm. In particular, cavitary lung cancer presented as having more solid nodules on CT images, and presented as more invasive in pathological findings. Interestingly, in their cohort, patients with cavitation had more frequent papillary histology, and none of them had micropapillary aspects.

Another possible implication of favorable histology implies the adoption of parenchymal saving strategies. Indeed, determining the feasibility of segmentectomy instead of lobectomy for small adenocarcinomas with mainly a solid component was the objective of two clinical trials, by Altorki and coworkers [16] and Saji and colleagues [17]. Accordingly, sublobar resection was not inferior to lobectomy for DFS, and OS was similar with the two procedures. It is important to consider the criteria defined in these studies when planning lung parenchyma sparing surgery, particularly whether the tumor meets dimensions < 2 cm. These are the only two randomized clinical trials that prove the feasibility of sublobar lung resection for a particular subgroup of NSCLC. In particular, for Altorki and coworkers [16], the pre-operative inclusion criteria were peripheral lung nodules ≤ 2 cm on a pre-operative CT scan and a presumption of lung cancer. The center of the tumor, seen on CT, had to be located in the outer third of the lung, in either the transverse, coronal or sagittal plane. Histology, in their protocol, was considered when performing risk stratification after surgery. Saji and colleagues [17] instead considered small-sized, invasive peripheral NSCLC (≤2 cm in diameter; consolidation-to-tumor ratio > 0.5; located in the outer third of the pulmonary parenchyma), with clinical stage IA NSCLC confirmed by thin-section CT. In their studies, conclusions were drawn on the basis of radiological and dimensional findings. The indications for LMPA should also be extended to include tumors with dimensions greater than 2 cm, with the possibility of performing segmentectomy in these cases as well.

LMPA’s features are defined after surgery, so they have not been used to drive treatment choices as of yet. It would be interesting to define the clinical factors predicting the presence of LMPA when it is suspected based on radiological findings (i.e., dimensions < 3 cm), in order to improve treatment strategies.

Some studies [18,19,20] have tried to better predict histological features, starting from radiological characteristics. Wang et al. [18], in their study, extracted the radiologic features of surgically resected < 3 cm adenocarcinomas to determine whether they were associated with ALK rearrangement. The analyses revealed that two clinicopathological and eight radiological features were significantly associated with ALK rearrangement status. These 10 features were subjected to statistical analysis, resulting in a single model containing three highly informative features associated with ALK rearrangement that included a lobulate margin, the absence of GGO, and a larger short axis of the largest lymph node (AUC 0.842 for specificity and sensitivity, CI 95%: 0.762–0.922).

Rong and coworkers [19] performed a study with the aim of evaluating high-resolution CT (HRCT) combined with PET/CT for pre-operative differentiation of invasive adenocarcinoma from preinvasive lesions and minimally invasive adenocarcinoma in the lungs with dimensions 3 cm or smaller. They concluded that the combined HRCT and PET/CT modality is an effective method to pre-operatively identify invasive adenocarcinoma for lung nodules smaller than 3 cm. The accuracy of this method in predicting invasive adenocarcinoma was 85.4% on a per-patient basis.

Aoki and colleagues [20] retrospectively analyzed the margin characteristics of nodules and the extent of ground-glass opacity (GGO) within nodules smaller than 3 cm on pre-operative thin-section CT, in order to predict histological features. According to their findings, coarse spiculation (*p* 0.01; OR, 3.26; 95% CI: 1.505, 7.068) and the extent of GGO (*p* 0.05; OR, 0.063; 95% CI: 0.004, 0.978) were predictive of regional lymph node metastasis.

Future radionics and metabolic studies might permit pre-operative identification of LMPA, guiding surgeons in choosing the appropriate lung resection, which could be sublobar and anatomic, such as segmentectomy.

Regarding recurrence, as previously stated, four recurrences were observed (28,5%), mainly locally (three), with only one occurring distally. To plan the best treatment, it could be of utmost importance to define criteria predicting the recurrence of LMPA. For example, another work from Wang and colleagues [21] analyzed the recurrence risk factors and clinical outcomes of peripheral pulmonary adenocarcinoma ≤ 3 cm. Accordingly, recurrence could be predicted based on sex, pathology and pleural invasion.

Given these premises, it would be interesting to analyze radiological and metabolic factors predicting the histology of LMPA and criteria predicting the risk of recurrence. This may be the starting point to validate the use of limited lung resections for these subgroups of adenocarcinoma, followed by an appropriate surveillance program. Indeed, with it being a subclassification that is still not understood in depth, precise indications regarding how to predict and treat this kind of adenocarcinoma are lacking. An important and interesting way to predict the presence of LMPA would be the identification of specific biomarkers in the peripheral blood (liquid biopsy). This would be helpful in pre-operative planning. In addition, in the literature, data on the application of liquid biopsy in this specific group are still lacking. This could be the topic of a future study on a prospective cohort in the frame of the LANTER project, which is currently ongoing in our institution [22], and has the precise aim of creating an avatar that is able to predict tumor features thought the analysis of multi-omics data, above all those extracted from pre-operative blood samples from early-stage lung cancer patients.

Our study presents some limitations. First of all, it is limited by its retrospective nature. Secondly, our analysis was performed on a small number of patients, which makes our results informative, but far from definitive. For this reason, we suggest future studies with a larger cohort. The presence of low-malignant-potential adenocarcinoma appeared to be, based on the results of the log-rank analysis, a good prognosis predictor in our cohort. Although our results are encouraging, further prospective studies on a larger cohort are needed to validate our findings.

## 5. Conclusions

Our study confirmed the prognostic value of the histological features of LMPA, with a low risk of recurrence and good CSS and DFS. The criteria for diagnosis are replicable and feasible for application. The clinical implications of these findings, such as pre-operative prediction and surveillance scheduling, may be the topic of future prospective studies.

## Figures and Tables

**Figure 1 curroncol-32-00217-f001:**
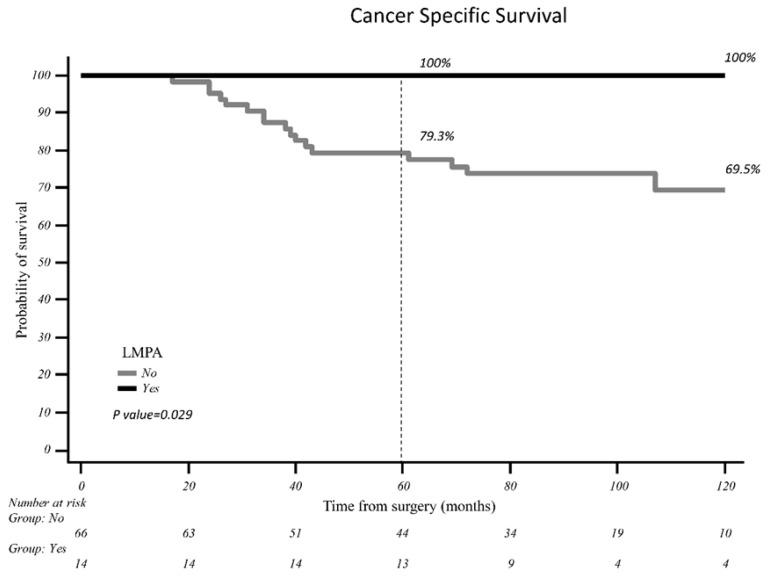
Log-rank analysis for LMPA and 10-year CSS.

**Figure 2 curroncol-32-00217-f002:**
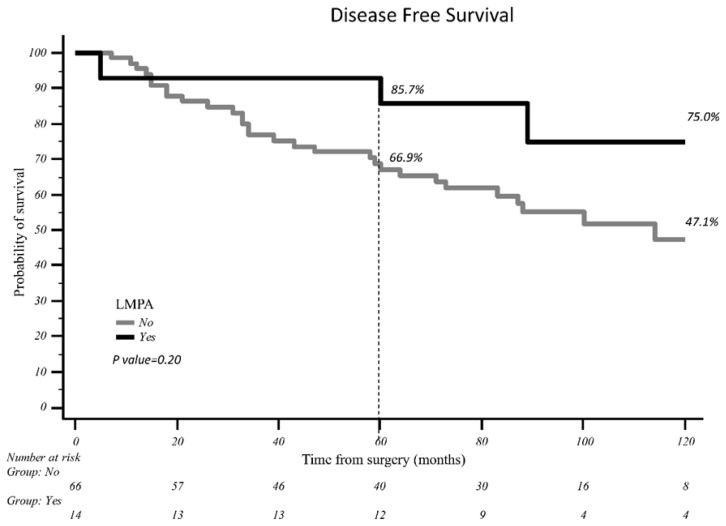
Log-rank analysis for LMPA and 10-year DFS.

**Table 1 curroncol-32-00217-t001:** Clinical and pathological features.

Clinicopathologic Findings	Total (N = 80)	LMPA (N = 14)
Age	72.9 (45–94)	73.1 (52–92)
Sex		
Female	35 (45)	5 (35.7)
Male	45 (55)	9 (64.3)
Tobacco exposure		
Yes	61 (76.3)	11 (78.6)
No	19 (23.7)	3 (21.4)
Side		
Right	46 (57.5)	
Left	34 (42.5)	
cStage (AJCC eighth edition)		
IA (pT1a)	16 (20)	6 (42.8)
IA2 (pT1b)	42 (52.5)	7 (50)
IA3 (pT2a)	22 (27.5)	1 (7.2)
pT dimension (cm)		
<1	16 (20)	
1–2	42 (52.5)	
2–3	22 (27.5)	
Resected lymph nodes		
^3^10	29 (36.3)	5 (28)
<10	51 (63.7)	9 (72)
Resected N2 stations		
^3^2	44 (55.3)	
<2	36 (44.7)	
Resected N2 lymph nodes		
^3^6	48 (60)	
<6	32 (40)	
LMPA	14 (17.5)	14 (100)

LMPA: low-malignant-potential adenocarcinoma, CSS: cancer-specific survival, DFS: disease-free survival, HR: hazard ratio.

**Table 2 curroncol-32-00217-t002:** Univariate analysis.

	Kind of Recurrence			*p*
	Local N (%)	Distant N (%)	Local and Distant N (%)	
Non-LMPA	19 (65.5%)	8 (27.6%)	2 (6.9%)	
LMPA	3 (75%)	1 (25%)	0	0.847

**Table 3 curroncol-32-00217-t003:** Analysis of recurrence.

	Kind of Recurrence			*p*
	Local N (%)	Distant N (%)	Local and Distant N (%)	
Non-LMPA	19 (65.5%)	8 (27.6%)	2 (6.9%)	
LMPA	3 (75%)	1 (25%)	0	0.847

## Data Availability

Data are available upon request to the corresponding author.

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
