# Peer review of "Low-Malignant-Potential Adenocarcinoma: A Histological Category with a Significantly Better Prognosis than Other Solid Adenocarcinomas at IA Stage"

_curroncol, 2025, doi:10.3390/curroncol32040217_

Round 1
Reviewer 1 Report
Comments and Suggestions for Authors
Review Comments for “Low Malignant Potential Adenocarcinomas: A Histological Category with a Significantly Better Prognosis Than Other Solid Adenocarcinomas at IA Stage”
Major Comments:
- Abstract and Introduction: The abstract presents the significance of classifying low malignant potential adenocarcinomas (LMPA) effectively. However, it would be beneficial to clarify the clinical implications of identifying LMPA in routine diagnostic practice. Additionally, providing a brief discussion on potential changes in patient management based on LMPA classification could strengthen the introduction.
- Results Interpretation: The findings suggest a significantly better prognosis for LMPA patients, but the sample size is a little bit small which will make the results not so solid. It would be helpful to acknowledge this limitation more explicitly and suggest future studies with a larger cohort.
- Discussion and Clinical Implications: The study successfully validates the prognostic value of LMPA, but a more in-depth discussion on how this classification could influence surgical decision-making or adjuvant therapy recommendations is needed. The potential application of molecular biomarkers in identifying LMPA preoperatively could also be explored.
Minor Comments:
- Line 31: “LMPA was not statistically significant for CSS and DFS” – The sentence could be revised for better clarity, e.g., “LMPA classification did not show a statistically significant impact on CSS and DFS, likely due to the limited number of events.”
- Grammatical Review: Several long and complex sentences could be broken down for better readability. For example, the sentence in Line 275-278 discussing Wang et al.’s radiological analysis could be made more concise. A thorough proofreading for clarity and conciseness is recommended.
Author Response
Major Comments:
Abstract and Introduction: The abstract presents the significance of classifying low malignant potential adenocarcinomas (LMPA) effectively. However, it would be beneficial to clarify the clinical implications of identifying LMPA in routine diagnostic practice. Additionally, providing a brief discussion on potential changes in patient management based on LMPA classification could strengthen the introduction.
Answer: thank you for this observation about abstract and introduction. They have both been implemented.
Results Interpretation: The findings suggest a significantly better prognosis for LMPA patients, but the sample size is a little bit small which will make the results not so solid. It would be helpful to acknowledge this limitation more explicitly and suggest future studies with a larger cohort.
Answer: thank you for this proper observation. A more explicit point to this limitation has been made
Discussion and Clinical Implications: The study successfully validates the prognostic value of LMPA, but a more in-depth discussion on how this classification could influence surgical decision-making or adjuvant therapy recommendations is needed. The potential application of molecular biomarkers in identifying LMPA preoperatively could also be explored.
Answer: thank you for this very interesting observation. We performed a little review of the literature, with the advanced tool of pubmed and the keywords “low malignant potential adenocarcinoma” “lung” and “biomarkers”, and actually, no results came out. This is the expression of the fact the it is a completely new field to be explored and this could be the aim of a future analysis in the frame of the LANTER project, going on in our institution. We implemented the text with possible clinical implications and future directions, under this specific point of view.
Minor Comments:
Line 31: “LMPA was not statistically significant for CSS and DFS” – The sentence could be revised for better clarity, e.g., “LMPA classification did not show a statistically significant impact on CSS and DFS, likely due to the limited number of events.”
Answer: thank you for this correction. The improvement has been added in the text.
Grammatical Review: Several long and complex sentences could be broken down for better readability. For example, the sentence in Line 275-278 discussing Wang et al.’s radiological analysis could be made more concise. A thorough proofreading for clarity and conciseness is recommended.
Answer: thank you for your insightful comment. Sentences and grammar have been revised, in particular English language has been revised by a mother tongue colleague.
Reviewer 2 Report
Comments and Suggestions for Authors
Congratulations to you on your successful histological study of LMPA and its association with a better prognosis. This is a novel finding and will offer a valuable contribution for the future clinical practice and reference. The entire manuscript was well written with acceptable language . However, there are no word spacing in many places found in the whole text. This should be thoroughly revised and amended before the editor can consider publication.
Author Response
Congratulations to you on your successful histological study of LMPA and its association with a better prognosis. This is a novel finding and will offer a valuable contribution for the future clinical practice and reference. The entire manuscript was well written with acceptable language. However, there are no word spacing in many places found in the whole text. This should be thoroughly revised and amended before the editor can consider publication.
Answer: thank you for this observation. Word spaces have been revised
Reviewer 3 Report
Comments and Suggestions for Authors
The study provides valuable insights into a rare tumor type that has not been extensively researched. Although the sample size is relatively small, this is understandable given the study's time frame and the low incidence of this tumor type. The discussion covers the broader context of the topic, and I have no objections to the study’s concept or its implications. However, the manuscript contains numerous technical and typographical errors that hinder readability.
To enhance clarity, I recommend the authors address the following points:
The language should be improved for better clarity.
There are several typographical errors and issues with spacing (both missing and excessive) throughout the text.
The abstract is somewhat confusing and difficult to read, primarily due to several unexplained acronyms and an overwhelming number of parameters. It lacks context and a clear statement of the study’s objectives. I would expect the abstract to convey the purpose of the research and the key findings, rather than attempting to summarize all the data.
The sentence outlining the study's aim at the end of the Introduction (lines 71-73) is unclear and should be reformulated.
Acronyms in Table 2 should be defined in the table legend.
It is unclear what data is referenced in line 206.
Comments on the Quality of English LanguageThe English could be improved to more clearly express the research.
Author Response
The study provides valuable insights into a rare tumor type that has not been extensively researched. Although the sample size is relatively small, this is understandable given the study's time frame and the low incidence of this tumor type. The discussion covers the broader context of the topic, and I have no objections to the study’s concept or its implications. However, the manuscript contains numerous technical and typographical errors that hinder readability.
To enhance clarity, I recommend the authors address the following points:
The language should be improved for better clarity.
There are several typographical errors and issues with spacing (both missing and excessive) throughout the text.
Answer: thank you for this important observation. Grammar, language and spacing have all been revised. In particular, English language has been revised by a mother tongue colleague.
The abstract is somewhat confusing and difficult to read, primarily due to several unexplained acronyms and an overwhelming number of parameters. It lacks context and a clear statement of the study’s objectives. I would expect the abstract to convey the purpose of the research and the key findings, rather than attempting to summarize all the data.
Answer: thank you for this insightful comment. As you correctly stated the abstract did not introduced appropriately the topic, so it has been revised.
The sentence outlining the study's aim at the end of the Introduction (lines 71-73) is unclear and should be reformulated.
Answer: thank you for this meaningful comment. The sentence about the study aims has been revised
Acronyms in Table 2 should be defined in the table legend.
Answer: thank you for this important observation, acronyms have been described
It is unclear what data is referenced in line 206.
Answer: thank you for this comment. The reference has been added in the text to make is clearer.